# Genome-wide association analysis of anthracnose resistance in the Yellow Bean Collection of Common Bean

**Kuwabo Kuwabo[1], Swivia M. Hamabwe[1], Paul Kachapulula[1], Karen Cichy[2], Travis Parker[3], Chikoti Mukuma[4], Kelvin Kamfwa[1]***

**1** Department of Plant Sciences, University of Zambia, Lusaka, Zambia, **2** USDA-ARS, Sugarbeet and Bean Research Unit, Bogue St., East Lansing, MI, United States of America, **3** Department of Plant Sciences/MS1, Section of Crop & Ecosystem Sciences, University of California, Davis, CA, United States of America, **4** Zambia Agricultural Research Institute, Kasama, Zambia

* kelvinkamfwa@gmail.com

**Data Availability Statement:** All relevant data are within the manuscript and its Supporting Information files.

## Abstract

Anthracnose caused by *Colletotrichum lindemuthianum* is a major disease of common bean (*Phaseolus vulgaris*) worldwide. Yellow beans are a major market class of common bean especially in eastern and southern Africa. The Yellow Bean Collection (YBC), which is comprised of 255 genotypes, and has not been used previously in genetic studies on anthracnose, is an excellent genetic resource for understanding the extent of anthracnose resistance and its genetic architecture in the yellow bean market class. The objectives of this study were i) evaluate the YBC for resistance to races 5, 19, 39, 51, 81, 183, 1050 and 1105 of *C. lindemuthianum*. and ii) conduct genome-wide association analysis to identify genomic regions and candidate genes associated with resistance to *C. lindemuthianum*. The YBC was genotyped with 72,866 SNPs, and genome-wide association analysis was conducted using Mixed Linear Model in TASSEL. Andean and Middle American genotypes with superior levels of resistance to the eight races were identified. YBC278 was the only one among 255 genotypes that was highly resistant to all eight races. Resistance to anthracnose in the YBC was controlled by major-effect loci on chromosomes Pv01, Pv03, Pv04, Pv05 and Pv07. The genomic region on Pv01, which overlapped with the Andean locus *Co-1* provided resistance to races 81, 1050 and 1105. Significant SNPs for resistance to race 39 were identified on Pv02. The genomic region on Pv04, which overlapped with known major-effect loci *Co-3*, *Co-15*, *Co-16*, *Co-y* and *Co-z*, provided resistance to races 5, 19, 51 and 183. Novel genomic regions for resistance to race 39 were identified on Pv05 and Pv07. Plant resistance genes (R genes) with NB-ARC and LRR domains, which occurred in clusters, were identified as positional candidate genes for genomic regions on Pv02 and Pv04.

## Introduction

Common bean (*Phaseolus vulgaris*) is a source of food and nutritional security for millions of households in Latin America and Africa [1]. Beans play an important role in combating

**Funding:** KK received scholarship from the Kirkhouse Trust of the United Kingdom. URL: https://www.kirkhousetrust.org The funders had no role in study design, data collection and analysis, decision to publish, or preparation of the manuscript.

**Competing interests:** The authors have declared that no competing interests exist.

malnutrition because of its richness in protein, carbohydrates and minerals (especially iron and zinc) [2].

Beans have a rich diversity in traits such as seed color, seed size and seed shape [3]. These traits have been used to classify beans in different market classes. One of the market classes is the yellow bean, which is comprised of both Andean and Middle American genotypes. Within the yellow bean market class, there are several seed types classifications based on the shade of yellow color. These include the Manteca, Canary (Canario), Mayocoba, and Njano classes [4,5]. Various yellow bean market classes are commercially important in Latin America, the Caribbean, and Africa [6,7]. In Africa, yellow beans are popular in Angola, Kenya, Tanzania and Zambia [7]. In Zambia, the Manteca yellow landrace known as Lusaka is preferred over other bean market classes and fetches higher market prices [8]. In Tanzania, the Njano class is the most popular yellow seed type and it is estimated that about 32% of total bean production there is for yellow beans, and mainly the Njano type [9]. Yellow beans have become increasingly important bean corridors of both East and Southern Africa [10].

Consumers prefer yellow beans to other market classes because of their superiority in several end-use traits. Manteca yellow beans have shorter cooking time and higher iron bioavailability than other market classes [11–14]. Despite the consumer preference and nutritional attributes of some yellow bean market classes have historically received less breeding efforts compared to other popular bean market classes.

Average yields for yellow bean are often lower than other bean classes. Several abiotic and biotic stresses contribute to low yields [15]. In some countries, continued use of yellow bean landraces with poor genetics and yield potential contribute to low yields. Diseases are a major contributing factor to the low yields of yellow beans. Anthracnose (ANTH) (caused by *Colletotrichum lindemuthianum*) is a major disease that affects yellow beans. Depending on varietal susceptibility and weather conditions (high humidity and moderate temperatures favors its spread), ANTH can cause yield losses of up to 100% [16]. ANTH is a seed borne disease and planting infected seed is a major mode of its transmission [17,18]. This transmission mode of ANTH makes its control/or management challenging because of the widespread use of on-farm saved seed in Africa and Latin America. Though ANTH can effectively be controlled using fungicides, smallholder farmers who are the majority of bean growers cannot afford fungicides. Additionally, there are health and safety concerns involved in using fungicides. Development and use of resistant varieties is the most cost-effective and ecologically sound management strategy for anthracnose.

ANTH resistance exists within the primary gene pool of common bean and several sources of resistance have been identified and used in breeding. Major-effect genes mainly control resistance to ANTH in common bean, though minor QTL have also been identified [19,20]. To date, 19 dominant major-effect ANTH resistance genes have been identified in common bean. These genes are classified as either Andean or Middle American depending on whether they were identified in an Andean or Middle American genotype. Andean genes include *Co-1*, *Co-12*, *Co-13*, *Co-14*, *Co-15*, *Co-x*, *Co-w*, *Co-y*, and *Co-z* while Middle American genes include *Co-2*, *Co-3*, *Co-4*, *Co-5*, *Co-6*, *Co-11*, *Co-16*, *Co-17*, *Co-u*, and *Co-v* [21–24]. The genes *Co-1*, *Co-3*, *Co-4* and *Co-5* are multi-allelic. Some of these major genes are located in genomic regions that contain clusters of disease resistance genes that control resistance to multiple diseases [25]. For example, *Co-1* is located in the genomic region (50,513,853–50,546,985 bp) that also contains *Pgh-1* [26], which is a major gene for resistance to angular leaf spot. In addition to the major genes, QTL with smaller effect on ANTH resistance have been reported [27].

The major challenge in breeding for durable ANTH resistance is the broad genetic diversity of *Colletotrichum lindemuthianum*. To date, over 298 races of *C. lindemuthianum* have been reported worldwide [28]. *C. lindemuthianum* races have co-evolved with the two gene pools of

common beans and are classified as either Andean of Middle American [21]. Andean races are virulent mostly on Andean beans while Middle American races are virulent on both Andean and Middle American beans. Because of the diversity of *C. lindemuthianum*, resistance often breaks down; therefore, there is need for continuous identification of new sources of resistance and understanding the genetic architecture of that resistance. Major genes for ANTH resistance follow gene-for-gene disease resistance concept. Therefore, some genes are effective against some races, but less effective against others. It is therefore important to identify genes that are effective against specific races so as to develop varieties with effective and durable resistance for a given geographic location. Diversity panels are important genetic resources that can serve as sources of resistance and also for understanding the genetic architecture of traits such as *C. lindemuthianum*. The Yellow Bean Collection (YBC) is one of the diversity panels that have been assembled in common bean [4]. The YBC is comprised of landraces, varieties and elite lines of yellow beans, which are variable for several traits including anthracnose resistance. The YBC, which has not been used previously in genetic studies on ANTH, is an excellent genetic resource for understanding the extent of ANTH resistance and its genetic architecture in the yellow bean market class.

QTL mapping using recombinant inbred lines and genome-wide association studies (GWAS) using diversity panels are two commonly used genetic approaches to understand the genetic architecture of traits. Both approaches have previously been used to identify genomic regions for resistance to variable races of *C. lindemuthianum* in common bean. One of the advantages of GWAS over QTL mapping is the higher mapping resolution that it offers, as GWAS takes advantage of historical recombination, leading to smaller linkage disequilibrium blocks. The enhanced mapping resolution of GWAS offers a better opportunity to precisely identify candidate genes for resistance to anthracnose. Additionally, GWAS using diversity panels provide an opportunity to identify minor effect QTL that QTL mapping populations using recombinant inbred lines cannot identify. Minor QTL are important sources of horizontal resistance, and if put in the same genetic background with major effect genes, which offers vertical resistance, they can confer durable resistance to multiple races of *C. lindemuthianum*. The objectives of this study were i) evaluate the yellow bean collection for resistance to eight races of *C. lindemuthianum*, and ii) conduct genome-wide association analysis to identify genomic regions and candidate genes associated with resistance to eight races of *C. lindemuthianum*.

## Materials and methods

### Plant materials

In the current study the Yellow Bean Collection (YBC) of 255 genotypes consisting of landraces, breeding lines and varieties with variable shades of yellow seed colors from Africa, North America, South America, Europe, Central America, Caribbean, Middle East and East Asia was used [4]. These shades of yellow seed colors include Amarillo dark and light, beige, brown, Canary (Mexican), Manteca, Mayocoba, and green-yellow (Njano). The YBC has both Andean and Middle American genotypes.

### Evaluation of the YBC for resistance to anthracnose

A total of eight races (5, 19, 39, 81, 183, 1050 and 1105) of *C. lindemuthianum* were used in the current study. These eight races were characterized from isolates collected from major bean-growing regions of Zambia using a set of 12 differential cultivars [29]. The eight races used in the current study offered a broad spectrum of virulence. Inoculation of the YBC followed a protocol described in Mungalu et al. [30]. This protocol involved planting the YBC (255

genotypes), anthracnose susceptible check (Kabulangeti) and anthracnose resistant check (G2333) on Styrofoam trays. A completely randomized design with three replications was used. Each replication had two seedlings in a well. Therefore, a total of six seedlings per genotype were evaluated for resistance to each race. Inoculation was conducted when seedlings had developed fully expanded unifoliate leaves. Evaluation of the YBC was conducted separately for each of the eight races.

**Inoculum preparation.** Inoculum preparation was conducted using a protocol described in Mungalu et al. [30]. Each race was cultured on petri dishes on either potato dextrose agar (PDA) 39 g $L^{-1}$ or modified Mathur's agar culture media made with dextrose (8 g $L^{-1}$), $MgSO_4.7H_2O$ (2.5 g $L^{-1}$), $KH_2PO_4$ (2.7 g $L^{-1}$), neopeptone (2.4 g $L^{-1)}$, yeast extract (2.0 g $L^{-1}$), and agar (16 g $L^{-1}$). The petri dishes were incubated in the dark at 23–25˚C for 7 to 10 days for sporulation. Sporulated plates where flooded with distilled water and allowed to settle for 20 minutes. Spores where then harvested by scrapping them off from the culture using a glass rod. The harvested spore suspension was then sieved through a double-layered cheesecloth into a clean beaker. The spore suspension concentration was then adjusted to 1.2 x $10^6$ spores per ml using a hemocytometer as described in Mungalu et al. [30]. Tween 20 was added to the inoculum for adhesion. Seedlings were thoroughly sprayed with the inoculum on both surfaces of the primary leaves and the stem using a hand sprayer. After inoculation, plants were left to air-dry for a few minutes before they were placed in a high humidity chamber (>90% humidity) for 72 hrs at room temperature. After this incubation seedlings were removed from the humidity chamber and transferred to greenhouse benches where they stayed for 5 days for further anthracnose development. Anthracnose severity on seedlings was rated based on a 1–9 CIAT scale [31]. The score range of 1–3 was considered as resistant and included plants with no visible symptoms or with few very small lesions, mostly on primary leaf veins, 4–6 was considered moderately resistant and included seedlings with small lesions on leaves and seedling stem, and 7–9 was considered susceptible and included dead seedlings and those with numerous small or enlarged lesions, with sunken cankers on leaves and seedling stem.

**Severity score analyses.** Statistical analyses on disease severity scores were conducted in SAS 9.3 [32]. Severity score data was first checked for normal distribution using PROC UNIVARIATE. Normality test results showed that severity scores for all races were normally distributed. Analysis of Variance was conducted using PROC MIXED following the mixed model:

$$Y_{ik} = \mu + \alpha_i + \gamma_k + E_{ik}$$

Where: $Y_{ik}$ was anthracnose severity score, with genotype i, replication k; αi was the fixed variable effect of the genotype i; γ was the random variable effect of a replication; ε was the residual associated with replication k in genotype i. γk was a random variable, which was assumed to be normally distributed with mean = 0.

## Genotypic data analyses

The YBC was genotyped with 72,866 Single Nucleotide Polymorphism (SNPs) markers using Genotyping by Sequencing (GBS). The DNA sequencing libraries were prepared using a single restriction enzyme ApeKI. The Genomics Core at Michigan State University sequenced the libraries using an Illumina standard HiSeq 4000 to generate 150 bp single-end reads. Additional details on genotyping can be found in Sadohara et al. [4]. The SNP data was used in population structure, kinship and SNP-disease severity analyses.

**Population structure.** Correction for population structure in a diversity panel such as the YBC is necessary to avoid false positives in GWAS. In the current study, the population

structure in the YBC was determined using Principal Component Analysis (PCA) in TASSEL v.5.0 [33]. The first five principal components, which together accounted for 62.5% of variation in the YBC, were used as covariates in the GWAS analyses.

**Kinship.** Kinship (cryptic relatedness) among genotypes in a diversity panel such as the YBC can result on false positive. In the current study, kinship among genotypes was investigated and corrected for in GWAS using the Identical by Descent method in TASSEL.

**Association analysis for anthracnose resistance.** SNPs significantly associated with resistance to races 5, 19, 39, 51, 81, 183, 1050 and 1105 were identified using the following Mixed Linear Model [34]:

$$Y = G + P + K + \varepsilon$$

Where Y the phenotype of a genotype; X was the fixed effect of the SNP; P was the fixed effect of population structure (from PCA matrix from TASSEL); K was the random effect of relative kinship (from kinship matrix from TASSEL); $\varepsilon$ was the error term, which was assumed to be normally distributed with mean = 0. MLM analysis was conducted in the software TASSEL v.5.0. Before use in MLM, SNP data was filtered for minor allele frequency (MAF = 5%), which reduced the number of usable SNPs for MLM to 55,000. The Bonferroni-corrected *P*-value of $1.0 \times 10^{-6}$ ($\alpha = 0.05$; 50,000 SNPs) was used as a threshold to determine the significance of the association between the SNP and severity score of a given race. The Manhattan plots for visualization of significant SNPs were made using a custom R script.

**Identification of candidate genes.** Candidate genes in genomic regions associated with resistance to the eight races of ANTH in the current study were identified from *Phaseolus vulgaris* v2.1 [35] in Phytozome using JBrowse following a previously described method [36]. Briefly, a gene was identified as a positional candidate gene based on two criteria: (i) it was within a genomic region of 400 kb of either upstream or downstream of the most significant SNP, and (ii) the functional role of a gene in disease resistance has been determined or proposed.

## Results

### Anthracnose severity score analysis

Highly significant (*P*<0.01) differences were observed among YBC genotypes in their reaction to races 5, 19, 39, 51, 81, 183, 1050 and 1105 of *C. lindemuthianum*. Severity scores for the YBC ranged from 1 to 9 for all eight races (Table 1).

The average severity score for the YBC to races 5, 19, 39, 51, 81, 183, 1050 and 1105 were 6.7, 6.3, 6.7, 5.2, 5.9, 6.4, 6.5 and 5.7, respectively (Table 1). The highest mean severity scores for the YBC were observed for Andean races 5 and 39. The frequency distribution of severity scores showed a bimodal distribution for all eight races (Fig 1).

The resistant and susceptible checks reacted to all eight races as expected. The resistant (G2333) and susceptible (Kabulangeti) checks consistently scored 1 and 9, respectively, for all eight races (Table 1). Of the 255 YBC genotypes evaluated, five genotypes (YBC278, YBC130, YBC173, YBC192 and YBC267) were highly resistant to most races (Table 2).

The genotype YBC278 was highly resistant (severity score < 1) to all eight races. The severity scores for all 255 YBC genotypes for the eight races are presented in S1 Table.

### Genome-wide association analysis

SNPs significantly associated with resistance to races 5, 19, 39, 51, 81, 183, 1050 and 1105 were identified on chromosomes Pv01, Pv02, Pv04, Pv05 and Pv07 (Figs 2–9; Table 3).

**Table 1. Means of anthracnose severity scores of 255 YBC accessions evaluated for resistance in the greenhouse.**

| ANTH race | Checks | | YBC | | |
|---|---|---|---|---|---|
| | G2333 | Kabulangeti | Mean | Range | ANOVA |
| Race 5 | 1.0 | 9.0 | 6.7±0.2 | 1.0–9.0 | *** |
| Race 19 | 1.0 | 9.0 | 6.3±0.21 | 1.0–9.0 | *** |
| Race 39 | 1.0 | 9.0 | 6.7±0.19 | 1.0–9.0 | *** |
| Race 51 | 1.0 | 9.0 | 5.2±0.21 | 1.0–9.0 | *** |
| Race 81 | 1.0 | 9.0 | 5.9±0.18 | 1.0–9.0 | *** |
| Race 183 | 1.0 | 9.0 | 6.4±0.2 | 1.0–9.0 | *** |
| Race 1050 | 1.0 | 9.0 | 6.5±0.19 | 1.0–9.0 | *** |
| Race 1105 | 1.0 | 9.0 | 5.7±0.21 | 1.0–9.0 | *** |

*** = Signifiacant ($p < 0.001$).

**Chromosome Pv01.** The genomic region spanning 49,151,104 bp—49,584,097 bp on chromosome Pv01 was significantly associated with resistance to race 81, 1050 and 1105 (Figs 2–4). The most significant SNP (Chr01_49584097) in this genomic region explained 41.2%, 27.5% and 21.4% of the YBC variation in ANTH severity caused by races 81, 1050 and 1105, respectively (Table 3).

**Chromosome Pv02.** Significant SNPs for resistance to the Andean ANTH race 39 were identified on Pv02 in a genomic region spanning from 49,155,240 bp—49,416,064 bp (Fig 5). The most significant SNP (Chr02_49318523) in this genomic region explained 30.1% of the YBC variation in ANTH severity caused by race 39 (Table 3).

**Chromosome Pv04.** Significant SNPs for races 19, 51 and 183 were identified in separate genomic regions of Pv04 (Figs 6–8). The first genomic region was at the beginning (1,067,693

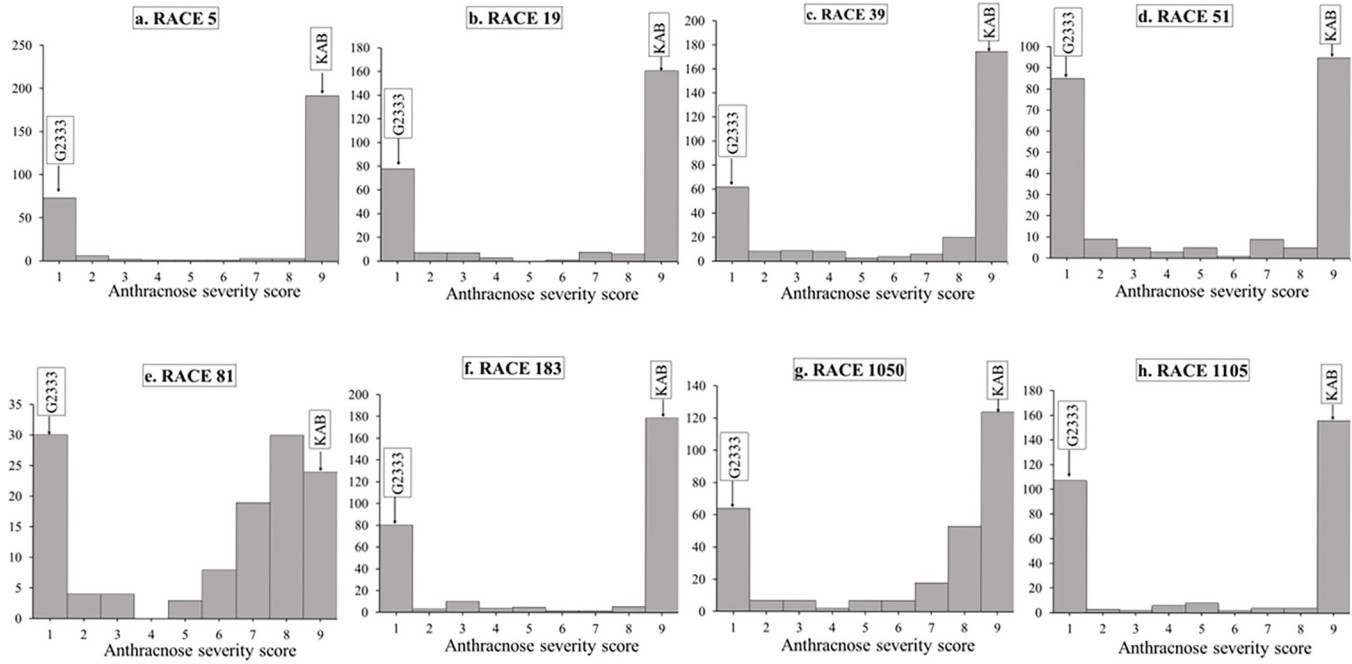

**Fig 1. Frequency distributions of severity scores for races 5, 19, 39, 51, 81, 183, 1050 and 1105 of *C. lindemuthianum* inoculated on the Yellow Bean Collection, resistant check (G2333) and susceptible check (Kabulangeti).**

**Table 2. Mean anthracnose severity scores for the five genotypes identified from the 255 Yellow Bean Collection genotypes as being highly resistant against races 5, 19, 39, 51, 81, 183, 1050 and 1105 of** *Colletotrichum lindemuthianum* **after artificial inoculation in the greenhouse at University of Zambia, Lusaka, Zambia.**

| YBC ID | ID | Country | Seed color | Genepool | Mean anthracnose scores | | | | | | | |
|--------|-----|---------|-----------|----------|--------|---------|---------|---------|---------|-------------|-------------|----------|
| | | | | | Race 5 | Race 19 | Race 39 | Race 51 | Race 81 | Race 183 | Race 1050 | Race 1105 |
| YBC278 | SMC28 | Uganda | Amarillo | Middle-American | 1.0 | 1.0 | 1.0 | 1.0 | 1.0 | 1.0 | 1.0 | 1.0 |
| YBC130 | P1527538 | Burundi | Green yellow | Andean | 1.0 | 1.0 | - | 1.2 | 1.0 | 1.0 | 1.0 | 1.0 |
| YBC173 | G22501 | Burundi | Green yellow | Andean | 1.0 | 1.0 | 9.0 | 1.0 | 1.0 | 1.0 | 1.3 | 1.0 |
| YBC267 | DAB933 | Uganda | Amarillo | Andean | 3.3 | 1.0 | 1.0 | 1.0 | 1.3 | 3.5 | - | 1.0 |
| YBC192 | Roba-1 | Uganda | Amarillo | Middle-American | 1.0 | 1.3 | - | 1.0 | 6.8 | 1.0 | 1.7 | 1.0 |

bp—2,140,558 bp) of Pv04. This genomic region was associated with resistance to races 19, 51 and 183. The $R^2$ for the most significant SNP (Chr04_2140558, Chr04_2140558 and Chr04_1067693) for races 19, 51 and 183 were 24.7%, 18.8% and 22.5%, respectively. The second genomic region identified on Pv04 was at 40,179,029 bp. This region was significantly associated with resistance to race 5 and the most significant SNP at this region explained 12.6% of the variation in the YBC for ANTH severity caused by race 5.

**Chromosome Pv05 and Pv07.** SNPs significantly associated with resistance to race 39 were identified on Pv05 and Pv07 (Fig 9). The most significant SNPs on Pv05 (Chr05_15211384; 15,211,384 bp) and Pv07 (Chr07_27213163; 27,213,163 bp) explained 13.3% and 14.2% of the variation in severity scores for race 39 (Table 3).

## Discussion

Anthracnose caused by *C. lindemuthianum* is a major disease of common bean. The productivity of yellow beans, a major market class in several African countries, is constrained by anthracnose. In the current study, we evaluated the YBC for its reaction to races 5, 19, 39, 51, 81, 183, 1050 and 1105 of *C. lindemuthianum*, and conducted GWAS to identify genomic regions and positional candidate genes conferring resistance to these eight races.

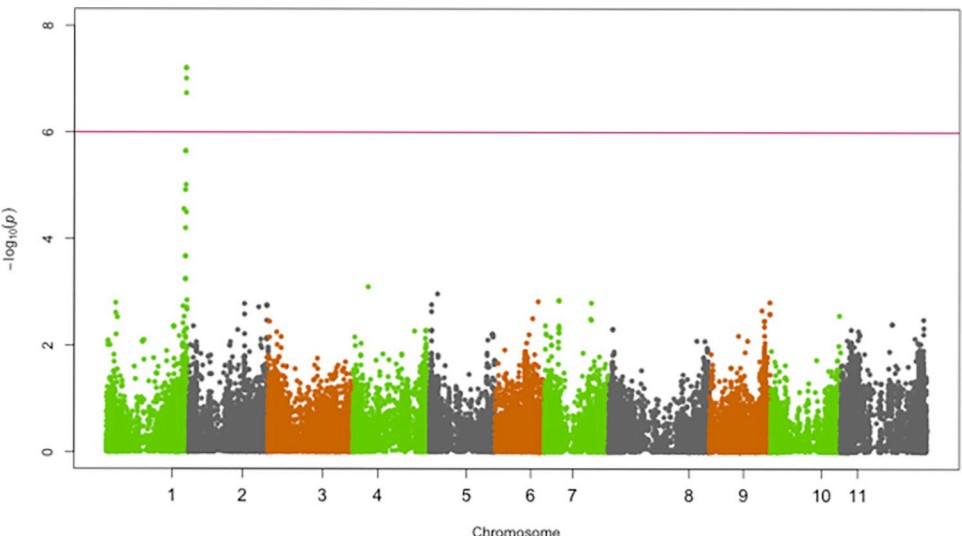

**Fig 2. Manhattan plot showing SNPs on chromosome Pv01 significantly associated with resistance to race 81 of the Yellow Bean Collection genotypes evaluated in the greenhouse at University of Zambia, Lusaka, Zambia.** The red solid horizontal line is the Bonferroni adjusted *P*-value (1.0E-06).

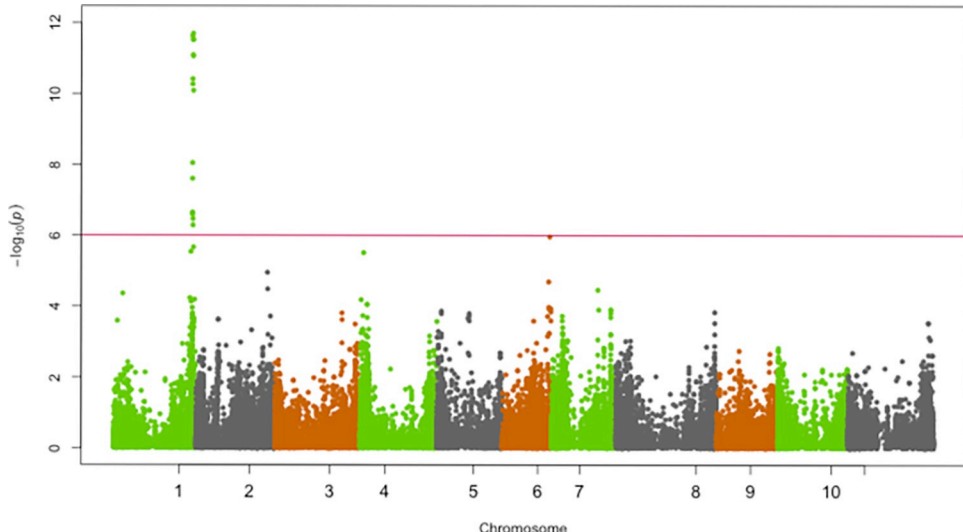

**Fig 3. Manhattan plot showing SNPs on chromosome Pv01 significantly associated with resistance to race 1050 of the Yellow Bean Collection genotypes evaluated in the greenhouse at University of Zambia, Lusaka, Zambia.** The red solid horizontal line is the Bonferroni adjusted *P*-value (1.0E-06).

The frequency distribution graphs for severity for all eight races showed a bimodal pattern, suggesting that major-effect genes were the basis of ANTH resistance, which is consistent with previous description of ANTH resistance in common bean as being controlled by major-effect genes. Identification of sources of resistance to anthracnose within the yellow bean market class would necessitate progress in the genetic enhancement of yellow beans for resistance to anthracnose. YBC278 was the only genotype in the YBC that was highly resistant (severity score of less than 1) to all the eight races used in the current study. YBC278, also known as SMC28, is a Middle American bean of race Mesoamerica from CIAT-Uganda. It has an

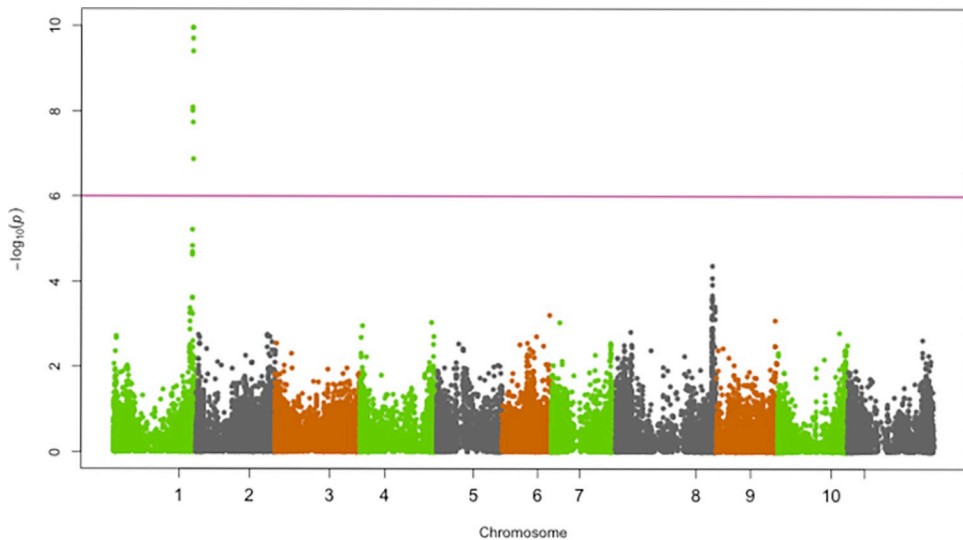

**Fig 4. Manhattan plot showing SNPs on chromosome Pv01 significantly associated with resistance to race 1105 of the Yellow Bean Collection genotypes evaluated in the greenhouse at University of Zambia, Lusaka, Zambia.** The red solid horizontal line is the Bonferroni adjusted *P*-value (1.0E-06).

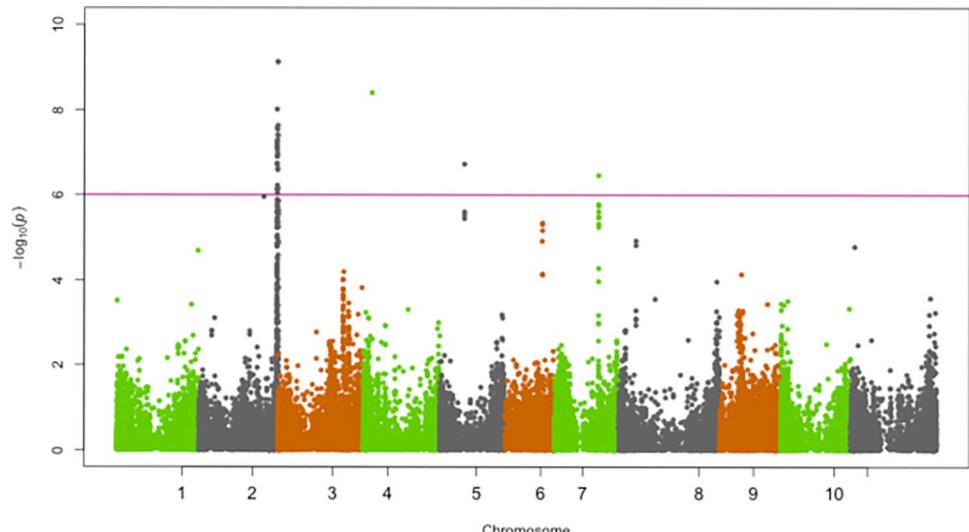

**Fig 5. Manhattan plot showing SNPs on chromosome Pv02, Pv04, Pv05 and Pv07 significantly associated with resistance to Andean ANTH race 39 of the Yellow Bean Collection genotypes evaluated in the greenhouse at University of Zambia, Lusaka, Zambia.** The red solid horizontal line is the Bonferroni adjusted *P*-value (1.0E-06).

Amarillo (dark yellow) seed type and a 100-seed weight of 27 grams. The identification of YBC278 as highly resistant is important, as it will serve as a source of superior resistance for developing yellow bean varieties with durable resistance to anthracnose. Genotypes YBC130 and YBC267 are the other two genotypes that showed superior levels of resistance to ANTH. YBC130 is an Andean landrace from Burundi with a green-yellow seedcoat and a 100-seed weight of approximately 43 grams. It was included in the Andean Diversity Panel as ADP0468 and is listed as PI 527538 in the USDA-NPGS gene bank. YBC267, also known as DAB 933, is an Andean bean from CIAT-Uganda of the Amarillo (dark) market category, with a 100-seed

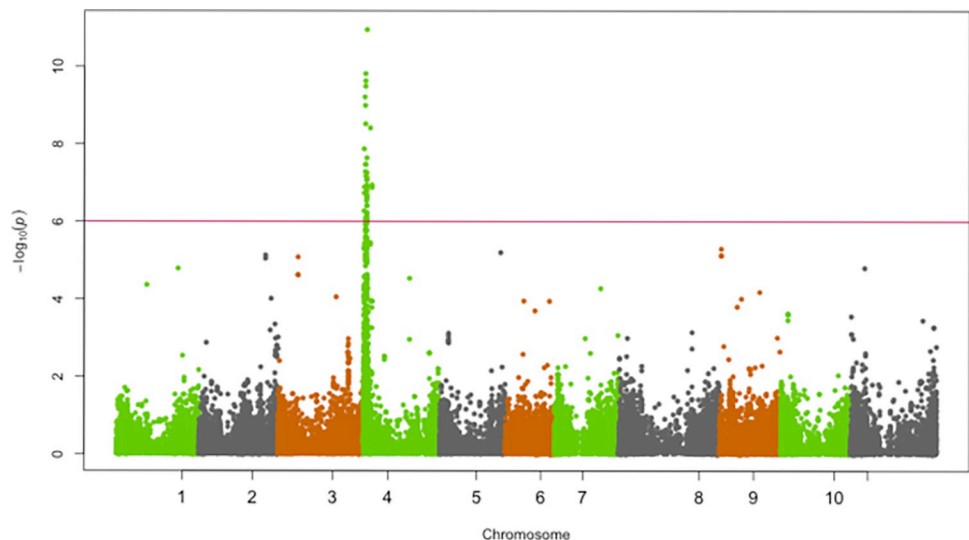

**Fig 6. Manhattan plot showing SNPs on chromosome Pv04 significantly associated with resistance to race 19, 51 and 183 of the Yellow Bean Collection genotypes evaluated in the greenhouse at University of Zambia, Lusaka, Zambia.** The red solid horizontal line is the Bonferroni adjusted *P*-value (1.0E-06).

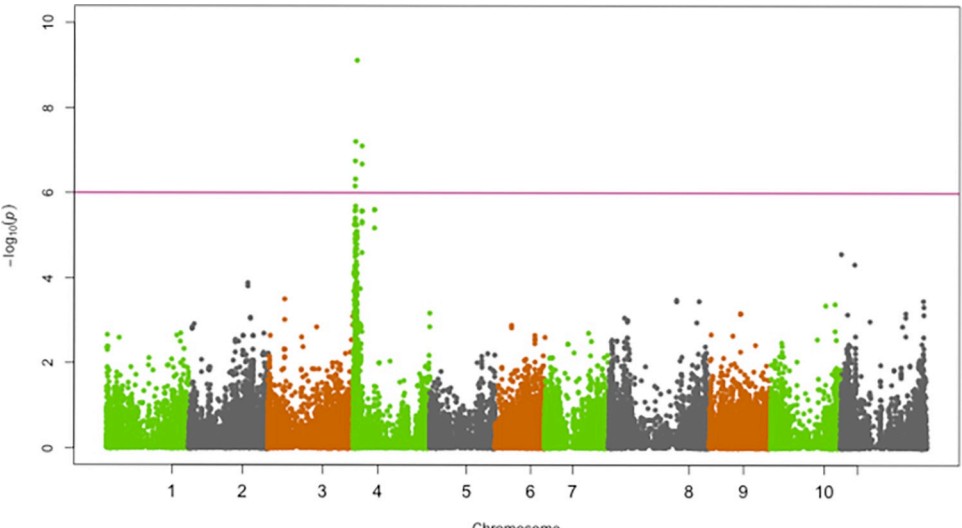

**Fig 7. Manhattan plot showing SNPs on chromosome Pv04 significantly associated with resistance to race 51 of the Yellow Bean Collection genotypes evaluated in the greenhouse at University of Zambia, Lusaka, Zambia.** The red solid horizontal line is the Bonferroni adjusted *P*-value (1.0E-06).

weight of 30 grams. These accessions could be used as alternative sources of ANTH resistance for yellow bean breeding. The highly resistant yellow bean accessions identified can be used as sources of anthracnose resistance in yellow bean breeding. This would help overcome the breeding challenges associated with poor recovery of yellow seed color in progenies from crosses of yellow bean with other colors.

The genetic basis of the observed resistance in the YBC to eight races of *C. lindemuthianum* was explored using genome-wide association analysis. Genomic regions associated with resistance to the eight races used in the current study were identified on chromosomes Pv01, Pv02, Pv04, Pv05 and Pv07.

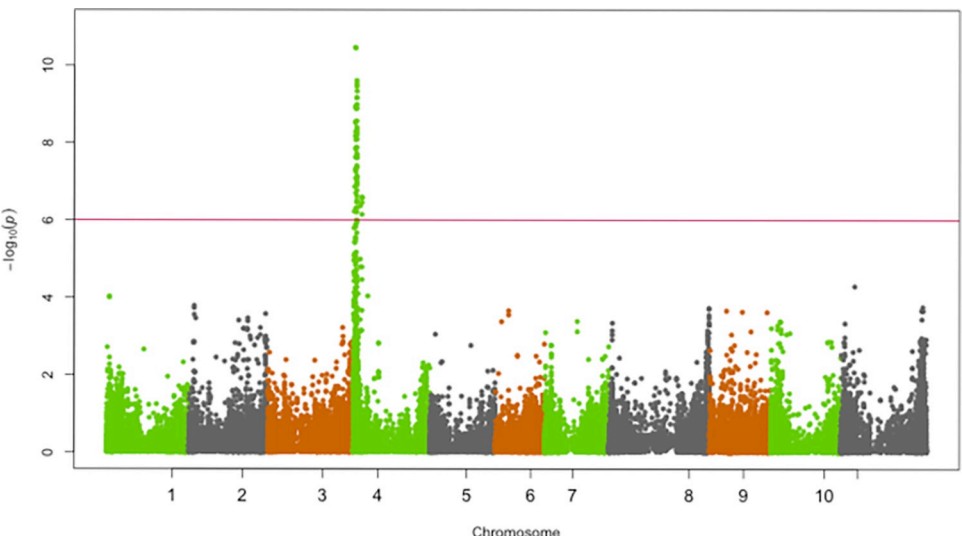

**Fig 8. Manhattan plot showing SNPs on chromosome Pv04 significantly associated with resistance to race 183 of the Yellow Bean Collection genotypes evaluated in the greenhouse at University of Zambia, Lusaka, Zambia.** The red solid horizontal line is the Bonferroni adjusted *P*-value (1.0E-06).

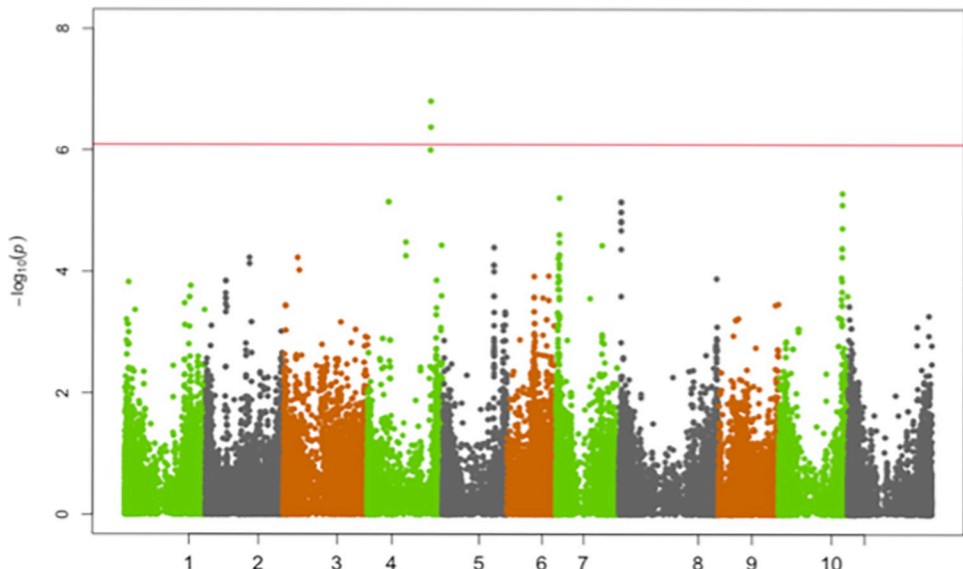

**Fig 9. Manhattan plot showing SNPs on chromosome Pv04 significantly associated with resistance to Andean ANTH race 5 in the Yellow Bean Collection evaluated in the greenhouse at University of Zambia, Lusaka, Zambia.** The red solid horizontal line is the Bonferroni adjusted *P*-value (1.0E-06).

Several SNPs within a 528 kbp region (49,151,104–49,679,631 bp) on Pv01 were significantly associated with resistance to race 81, and the highly virulent races 1050 and 1105. This is a major-effect QTL and it appears to be the major locus *Co-1*, which confers resistance mainly to Middle American races of *C. lindemuthianum*. The SNPs most significantly associated with resistance to races 81, 1050, and 1105 were found at positions 49,584,097, 49,583,965, and 49,584,097 of Pv01, respectively. In each case, the most SNP most significantly associated with ANTH resistance is in the gene model Phvul.001G243800, a protein-kinase recently proposed to underlie *Co-1* [37,38]. This is a powerful validation of the role of this gene, as well as the experimental methods employed in this study. Previous studies have shown existence of both Andean and Middle American races in East and Southern African countries including Zambia, which underscores the importance of having *Co-1* gene in Andean genotypes such as the yellow beans.

**Table 3. SNP markers significantly associated with resistance to *Colletotrichum lindemuthianum* in the Yellow Bean Collection.**

| Trait | Chr | SNP | Position (bp) | *P*-value | *R²* (%) |
|---|---|---|---|---|---|
| Race 5 | Pv04 | Chr04_40179029 | 4,0179,029 | 1.5E-06 | 12.6 |
| Race 19 | Pv04 | Chr04_2140558 | 2,140,558 | 1.1E-11 | 24.7 |
| Race 39 | Pv02 | Chr02_49318523 | 49,318,523 | 7.4E-10 | 30.1 |
| Race 39 | Pv05 | Chr05_15211384 | 15,211,384 | 1.19E-07 | 13.3 |
| Race 39 | Pv07 | Chr07_27213163 | 27,213,163 | 7.15E-07 | 14.2 |
| Race 51 | Pv04 | Chr04_2140558 | 2,140,558 | 7.6E-10 | 18.8 |
| Race 81 | Pv01 | Chr01_49584097 | 49,584,097 | 6.2E-08 | 41.2 |
| Race 183 | Pv04 | Chr04_1067693 | 1,067,693 | 3.4E-11 | 22.5 |
| Race 1050 | Pv01 | Chr01_49583965 | 49,583,965 | 1.9E-12 | 27.5 |
| Race 1105 | Pv01 | Chr01_49584097 | 49,584,097 | 1.1E-10 | 21.4 |

Chr = Chromosome; bp = base pairs; *P*-value = level of significance; E = exponential; *R²* = proportion of variation of anthracnose resistance explained by the significant SNP.

The genomic region 49,318,523 bp on Pv02 contained SNPs that were significantly associated with resistance to races 19 and 39. The identification of YBC genotypes resistant to races 19 and 39, and their associated resistance genomic regions has breeding significance to Zambia and other bean producing countries for two reasons. First, race 19 is the most prevalent race in the bean-producing regions of Zambia. Second, race 39 is a highly virulent Andean race [27] and it is important that yellow bean genotypes with superior resistance to race 39 are identified to serve as source of resistance for genetic improvement of yellow beans in countries where it is a major market class. The association between race 39 and the genomic region on Pv02 (49,318,523 bp) has previously been reported [27]. Eight plant disease resistance genes (R genes) including Phvul.002G323100, Phvul.002G323200, Phvul.002G323300, Phvul.002G323400, Phvul.002G323404, Phvul.002G323704, Phvul.002G323708, and Phvul.002G323712 with NB-ARC and LRR domains were identified as positional candidate genes for resistance to race 39 identified on Pv02. R genes encode receptors that play a key role in the recognition of avirulence (Avr) products expressed by the pathogen during infection, which triggers a defense response [39], and is the basis for gene-for-gene or race-specific resistance for ANTH observed in common bean [40].

The genomic region on 1.1 Mbp– 2.1 Mbp on Pv04 conferred resistance to races 19, 51 and 183. This genomic region overlaps with major-effect loci *Co-3*, *Co-15*, *Co-16*, *Co-y* and *Co-z*. The relatively large number of races whose resistance was controlled by this genomic region is indicative of its significant role in the YBC as a source of broad-spectrum resistance to multiple races of *C. lindemuthianum*. The genomic region 1.1 Mbp– 2.1 Mbp identified in the current study as providing resistance to races 19, 51 and 183 was previously reported to also provide resistance to races 7, 19, 49, 55, 109, 530, 566 and 1331 [27,30], which underscores the importance of this genomic region for resistance to multiple races of anthracnose. Five disease resistance (R) genes including Phvul.004G015600, Phvul.004G015800, Phvul.004G015900, Phvul.004G016000 and Phvul.004G016532 with the NB-ARC and LRR domains were identified as candidate genes for resistance to races 19, 51 and 183 at the 1.1 Mbp—2.1 Mbp genomic region on Pv04. These R genes occurred in a cluster and it is possible that some of them provide race-specific resistance because this genomic region provided resistance to a relatively large number of races. Given that *Co-3* has previously been reported as multi-allelic it is possible that resistance to races 19, 51 and 183 reported in the current study may be conditioned by different alleles. In addition to overlapping with *Co-3*, the genomic region on Pv04 identified in the current study also overlaps with the telomere cluster of resistance genes for rust (*Ur-5* and *Ur-14*) [41], angular leaf spot (*Phg-3*) [41], powdery mildew [42] and halo blight QTL (HB4.2) [43], which are major diseases of beans in Africa. Given the convergence of resistance to several races of ANTH and other major diseases on Pv04 (1.1 Mbp– 2.1 Mbp), it is a worthwhile target for Marker-assisted selection to develop common bean varieties with durable resistance to ANTH and also multiple resistances to ANTH, rust and angular leaf spot. The current study has identified another genomic region on Pv04 to control resistance to ANTH. This genomic region located at the distal end on Pv04 (40.2 Mbp) is a major locus ($R^2$ = 12.6%) and the basis for resistance in the YBC to race 5 (Fig 5). This is the first report of ANTH resistance to race 5 at this genomic region.

Significant SNPs for resistance to race 39 were identified on Pv05 (15.2 Mbp) and Pv07 (27.2 Mbp). This is the first report of resistance for race 39 on Pv05 and Pv07. Banoo et al. [20] identified significant SNPs for resistance to races 87 and 73 on Pv05 and Pv07, but in a genomic region different from the one identified for race 39 in the current study. Resistance to race 39 has previously been reported on other chromosomes but not in the genomic regions on Pv05 and Pv05 identified in the current study. The locus *Co-2* on Pv11 was previously identified using a population of RILs as the source of resistance to race 39 [26]. A previous GWAS

conducted using the ADP reported resistance to race 39 only on chromosome Pv04 [27], the same region we have also identified in the current study. Our study has not only identified a previously known genomic region on Pv04 as source of resistance to race 39, but has also identified novel major QTL on Pv05 and Pv07 as additional genomic regions for resistance to race 39.

## Conclusion

Three YBC accessions with superior resistance to races 5, 19, 51, 81, 183, 1050 and 1105 were identified. The accession YBC278 was notable because it is the only one in the YBC that showed strong resistance to all eight races of *C. lindemuthianum* used in the current study.

The genetic architecture for resistance to eight races of *C. lindemuthianum* in the YBC has been determined. Major-effect loci on Pv01 (*Co-1*), Pv02 and Pv04 (which overlaps with *Co-3*, *Co-13*, *Co-15*, *Co-y* and *Co-z*) controlled most of the observed resistance to eight races observed in the YBC. In addition to the identified major-effect loci genomic regions on Pv01, Pv02 and Pv04, novel genomic regions on Pv04 (40.2 Mbp), Pv05 (15.2 Mbp) and Pv07 (27.2 Mbp) were significantly associated with the Andean races 5 and 39. Clusters of R genes with NB-ARC and LRR domains on Pv02 and Pv04 were identified as candidate genes.

## Supporting information

**S1 Table. Severity scores for races 5, 19, 39, 51, 81, 183, 1050 and 1105 of *C. lindemuthianum* for the 255 Yellow Bean Collection genotypes evaluated in the greenhouse at University of Zambia, Lusaka, Zambia.**
(XLSX)

## Acknowledgments

We thank the University of Zambia Bean Breeding Program team for their assistance in conducting inoculations. We also thank Sadohara of Dr. Karen Cichy's for DNA collection.

## Author Contributions

**Conceptualization:** Kuwabo Kuwabo, Kelvin Kamfwa.

**Formal analysis:** Kuwabo Kuwabo, Swivia M. Hamabwe, Travis Parker, Chikoti Mukuma, Kelvin Kamfwa.

**Funding acquisition:** Kelvin Kamfwa.

**Methodology:** Swivia M. Hamabwe, Karen Cichy, Travis Parker, Chikoti Mukuma.

**Resources:** Karen Cichy.

**Supervision:** Paul Kachapulula, Kelvin Kamfwa.

**Writing – original draft:** Kuwabo Kuwabo, Swivia M. Hamabwe, Karen Cichy, Travis Parker, Chikoti Mukuma, Kelvin Kamfwa.

**Writing – review & editing:** Kuwabo Kuwabo, Swivia M. Hamabwe, Karen Cichy, Travis Parker, Chikoti Mukuma, Kelvin Kamfwa.

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
