## [Decision Letter · Decision Letter 0]

25 Jun 2023

PONE-D-23-01331Genome-wide association analysis of anthracnose resistance in the Yellow Bean Collection of Common BeanPLOS ONE

Dear Dr. Kelvin,

Thank you for submitting your manuscript to PLOS ONE. After careful consideration, we feel that it has merit but does not fully meet PLOS ONE’s publication criteria as it currently stands. Therefore, we invite you to submit a revised version of the manuscript that addresses the points raised during the review process.

ACADEMIC EDITOR:  I suggest authors modify it according to the reviewer's comments. 

We look forward to receiving your revised manuscript.

Kind regards,

Karthikeyan Adhimoolam

Academic Editor

PLOS ONE

“The Kirkhouse Trust of the United Kingdom through the African Bean Consortium Project provided scholarship to Mr. Kuwabo Kuwabo. The opinions expressed in this publication are those of the authors and do not necessarily reflect the views of the Kirkhouse Trust.”

“KK received scholarship from the Kirkhouse Trust of the United Kingdom. URL:https://www.kirkhousetrust.org

The funders had no role in study design, data collection and analysis, decision to publish, or preparation of the manuscript”

Reviewers' comments:

Reviewer's Responses to Questions

**Comments to the Author**

1. Is the manuscript technically sound, and do the data support the conclusions?

Reviewer #1: Partly

Reviewer #2: Yes

2. Has the statistical analysis been performed appropriately and rigorously? 

Reviewer #1: No

Reviewer #2: Yes

3. Have the authors made all data underlying the findings in their manuscript fully available?

Reviewer #1: No

Reviewer #2: Yes

4. Is the manuscript presented in an intelligible fashion and written in standard English?

Reviewer #1: No

Reviewer #2: Yes

5. Review Comments to the Author

Reviewer #1: 1. The work done in this study refers to a very important biotic constraint in common bean production and some useful information has been generated in this study but authors have failed to signify the importance and novelty of this work as the MS has been written as routine presentation of the data generated in the study.

2. The methodology details of this work are detailed in reference 4 (Genet Resour Crop Evol. 2022; 69(4): 1627-1648) cited in this study. Only citation of reference to qualify the methodology used is not sufficient to help the reader and signify the work done, hence a gist o the protocol or procedures used need mention in the material and method section.

3. The presentation of the data in figures is very complex and is not able to access the descriptions of the results, hence need proper attention to depict the results in figures.

4. Almost more than 95% of the genotypes were susceptible to most of the races. it's not clear that what were the basis of selecting the panel of genotype in this study need clarity.

5. There are few more observations in the reviewed copy of the MS, which can improve the quality of the work done for better resolution of the results and facilitate the reader to come to conclusion of the study.

6. The results are not properly described. There is no mention of the SNPs that can help in future study and likewise other output issues.

7. The MS needs major revision with a view to highlight the novelty of the study, methodology followed and results section in particular.

Reviewer #2: The manuscript is well written and finding a few novel anthracnose associated SNPs on PV 05 and Pv07 are of interest. There are a few typos that need due attention and are listed below. A few important references are missing from the manuscript.

L68-69. There is typo. Delete of and insert, hey?

L86. ANTH may be replaced with anthracnose

L89-91. Recently Nabi et al., 2022 have compiled this information. Not few QRLs now many have been identified do check (Simson et al., 2022; Banoo et al., 2020).

L111-112. Please do rewrite it.

L117. Yellow bean Collection (YBC). Do write this acronym before also and define it in the first para., L144,

L132-134. Do rewrite it

L135. Delete effect.

L155 and L166 [28] may be replaced with Mungalu et al., (2020) as author year style.

L168-169. Sub-script the magnesium sulphate and potassium dihydrogen othrophosphate

L176 do provide reference at the end

L206 and 212 [4] write the reference in author (year) style

L435. Do add a recent reference (Lima et al., 2023) that validated this gene model for anthracnose resistance. There are numerous GWAS studies that indicate association of this region with multiple anthracnose resistance. Do add write of such reference to support your conclusion

L467. Add a few more references

L492. Banoo et al also found SNP on PV05 and 07 against race 87 and 73. Do add that also.

6. PLOS authors have the option to publish the peer review history of their article (what does this mean?). If published, this will include your full peer review and any attached files.

Reviewer #1: No

Reviewer #2: No

---

## [Author Response · Author response to Decision Letter 0]

11 Aug 2023

Responses to Reviewer Comments 

Reviewer # 1

Comment: The work done in this study refers to a very important biotic constraint in common bean production and some useful information has been generated in this study but authors have failed to signify the importance and novelty of this work as the MS resistance has been written as routine presentation of the data generated in the study.

Response: This is the first Genome-wide Association Study (GWAS) on common bean anthracnose conducted using the yellow bean diversity panel. The study has provided important and novel information on the genetic architecture of anthracnose resistance in yellow beans, which is a major market class of common bean. In addition to confirming previously identified genomic regions for anthracnose (ANTH) resistance, the study has reported novel genomic regions for ANTH resistance. The genomic regions on chromosomes Pv05 and Pv07 that provided resistance to race 39 are novel and reported for the first time in this manuscript. The other novelty is in the races used in the current study. For the majority of these races, it is the first time the genomic regions providing resistance to them has been established, and clusters of resistance genes associated with these specific races identified. Furthermore, this study has identified a yellow bean genotype YBC278 that is highly resistant to ANTH. This is a key finding with potentially major impact on the bean breeding community working on developing yellow bean varieties resistant to ANTH. There is no previous report of a yellow bean genotype that has higher ANTH resistance than YBC278 identified in the current study. In addition to the “standard” presentation of GWAS results in the form of Manhattan plots and tables showing genomic regions associated with resistance used in the current, the manuscript has gone further to better understand the positional candidate genes underlying the significant genomic regions. The study has determined that some of the positional candidate genes underlying the identified resistance loci occur in clusters, which could possibly explain why some of these resistance loci provide resistance to multiple races of C. lindemuthianum.

Comment: The methodology details of this work are detailed in reference 4 (Genet Resour Crop Evol. 2022; 69(4): 1627-1648) cited in this study. Only citation of reference to qualify the methodology used is not sufficient to help the reader and signify the work done, hence a gist of the protocol or procedures used need mention in the material and method section.

Response: We have revised the sections, Plant materials, Genotypic data analyses and Population structure under Materials and Methods that hopefully addresses this comment. 

Comment: The presentation of the data in figures is very complex and is not able to access the descriptions of the results, hence need proper attention to depict the results in figures.

Response: Results for this GWAS have been presented in a “standard” way, which involves use of histograms and tables to summarize the phenotypic data, and identify patterns in the data. Further, GWAS results have been presented using Manhattan plots and tables to show genomic regions associated with resistance to the eight races used in the current study. These tables, histograms and Manhattan plots were used to ensure that presentations of results are not complicated, and followed a “standard” way of presenting GWAS results.

Comment: Almost more than 95% of the genotypes were susceptible to most of the races. it's not clear that what were the basis of selecting the panel of genotype in this study need clarity.

Response: A number of considerations went into assembling the panel and its choice for use in the current study. First, the panel was assembled to specifically focus on yellow beans. The second consideration was that the panel had to have rich diversity of yellow bean genotypes and this was actualized by ensuring that the panel was comprised of yellow bean genotypes from breeding programs from different parts of the world. Yes, 95% of materials might have been susceptible to most races, but genotypes (few) with superior resistance (Table 2) comparable to that of even the resistant check for some races were identified. This demonstrated the importance and novelty of the study, because the identified resistant genotypes, few as they may be, are crucially important as novel sources of resistance to breed for improved ANTH resistance in yellow bean.

Comment: There are few more observations in the reviewed copy of the MS, which can improve the quality of the work done for better resolution of the results and facilitate the reader to come to conclusion of the study.

Response: Thank you for the observations. The manuscript has been revised to include all the observations that were in the reviewed copy of the manuscript.

Comment: The results are not properly described. There is no mention of the SNPs that can help in future study and likewise other output issues.

Response: We have revised the Results section to include the name of the SNPs associated with resistance to races used in the current study. The SNPs are mentioned on pages 17, 18 and 19 under Results. Furthermore, the SNP names and positions have been provided in Table 3. The reason we did not provide the SNP names initially was because we thought presenting positions or genomic regions significantly associated with resistance was more useful and could facilitate future comparisons of genomic regions identified in the current study with future studies. Names of SNPs do change depending on the genotyping platform used, but the positions do not change or change very slightly when there is a new version of the genome.

Reviewer # 2

Comment: The manuscript is well written and finding a few novel anthracnose associated SNPs on PV 05 and Pv07 are of interest. 

Response: Thank you.

Comment: There are a few typos that need due attention and are listed below

Response: The typo’s have been corrected.

Comment: A few important references are missing from the manuscript.

Response: Several missing key references have been added to the revised manuscript.

Comment: L68-69. There is typo. Delete of and insert, hey?

Response: It was a typo and it has been corrected.

Comment: L86. ANTH may be replaced with anthracnose

Response: Replaced.

Comment: L89-91. Recently Nabi et al., 2022 have compiled this information. Not few QRLs now many have been identified do check (Simson et al., 2022; Banoo et al., 2020).

Response: Corrected and the three references have been added.

Comment: L111-112. Please do rewrite it.

Response: The sentence has been rewritten. 

Comment: L117. Yellow bean Collection (YBC). Do write this acronym before also and define it in the first paragraph.

Response: Corrected

Comment: L132-134. Do rewrite it

Response: The sentence has been rewritten. 

Comment: L135. Delete effect.

Response: Corrected.

Comment: L155 and L166 [28] may be replaced with Mungalu et al., (2020) as author year style.

Response: Corrected.

Comment: L168-169. Sub-script the magnesium sulphate and potassium dihydrogen othrophosphate

Response: Corrected.

Comment: L176 do provide reference at the end

Response: The reference has been added.

Comment: L206 and 212 [4] write the reference in author (year) style

Response: The author style has been changed.

Comment: L435. Do add a recent reference (Lima et al., 2023) that validated this gene model for anthracnose resistance. There are numerous GWAS studies that indicate association of this region with multiple anthracnose resistance. Do add write of such reference to support your conclusion

Response: The suggested reference has been added.

Comment: L467. Add a few more references

Response: References have been added.

Comment: L492. Banoo et al also found SNP on PV05 and 07 against race 87 and 73. Do add that also.

Response: The suggested reference has been added.

---

## [Decision Letter · Decision Letter 1]

6 Sep 2023

PONE-D-23-01331R1Genome-wide association analysis of anthracnose resistance in the Yellow Bean Collection of Common BeanPLOS ONE

Dear Dr. Kamfwa 

Thank you for submitting your manuscript to PLOS ONE. After careful consideration, we feel that it has merit but does not fully meet PLOS ONE’s publication criteria as it currently stands. Therefore, we invite you to submit a revised version of the manuscript that addresses the points raised during the review process.

ACADEMIC EDITOR: I request that the authors make the necessary corrections following the reviewer's comments. 

We look forward to receiving your revised manuscript.

Kind regards,

Karthikeyan Adhimoolam

Academic Editor

PLOS ONE

Journal Requirements:

Reviewers' comments:

Reviewer's Responses to Questions

**Comments to the Author**

1. If the authors have adequately addressed your comments raised in a previous round of review and you feel that this manuscript is now acceptable for publication, you may indicate that here to bypass the “Comments to the Author” section, enter your conflict of interest statement in the “Confidential to Editor” section, and submit your "Accept" recommendation.

Reviewer #1: All comments have been addressed

Reviewer #2: All comments have been addressed

2. Is the manuscript technically sound, and do the data support the conclusions?

Reviewer #1: Yes

Reviewer #2: Yes

3. Has the statistical analysis been performed appropriately and rigorously? 

Reviewer #1: Yes

Reviewer #2: Yes

4. Have the authors made all data underlying the findings in their manuscript fully available?

Reviewer #1: Yes

Reviewer #2: Yes

5. Is the manuscript presented in an intelligible fashion and written in standard English?

Reviewer #1: Yes

Reviewer #2: Yes

6. Review Comments to the Author

Reviewer #1: 1. The authors have made very good attempt to impove the MS and now seems to be in order, however, still there are some queries and observations that need quick attention so that the MS could be processed further.

2. Page 8, line182-186, Dear Author, see the ref 34, as per this plant showing reaction 1-3 are considered resistant, whereas plant showing reaction 4-9 are considered as susceptible, so please clarify how you have rated score 4-6 as MR, need proper explanation.

3. Page 28, line no 414-415, check the statement and correct

4. There are some typographical and grammatical errors marked in the reviewed copy, be taken care of to improve the MS quality

Reviewer #2: There are a few typos that need due attention,

L98-99. Please check the sentencse -----genomic?. Position or region must be included

L 104. There are about 298 races now reported in C. lindemuthianum (Nunes et al., 2021), 182 races were previously identified (Padder et al., 2017).

L132. Delete “GWAS using”

L175. The author (year) is included but do check Plosone reference style in such a case.

L301. Delete one that. That is written twice

L385. (Figu) or (Fig.)

L494. Banoo et al., 2020. Please see reference author (year) style of Plosone.

7. PLOS authors have the option to publish the peer review history of their article (what does this mean?). If published, this will include your full peer review and any attached files.

Reviewer #1: No

Reviewer #2: **Yes: **Bilal A Padder

---

## [Author Response · Author response to Decision Letter 1]

9 Sep 2023

Responses to Reviewer Comments 

Reviewer # 1

Comments: The authors have made very good attempt to improve the MS and now seems to be in order, however, still there are some queries and observations that need quick attention so that the MS could be processed further.

Response: The queries and observations have been worked on.

Comment: The author must clearly state that this is a unique panel not used earlier, need proper mention.

Response: A sentence has been included in both the abstract and introduction that the Yellow Bean Collection has not been previously used in genetic studies.

Comment: Page 8, line182-186, Dear Author, see the ref 34, as per this plant showing reaction 1-3 are considered resistant, whereas plant showing reaction 4-9 are considered as susceptible, so please clarify how you have rated score 4-6 as MR, need proper explanation.

Response: The severity scores in ref 34 (Balardin et al.) were on a scale 1-9. The reason they classified 4-9 as susceptible was because that paper was on characterization of C. lindemuthianum, and a final binary score of either resistant or susceptible is required. However, in breeding we have three classifications (resistant, moderately resistant and susceptible). Scores of 4-6 are classified as moderately resistant. We have replaced Balardin et al. with a more appropriate citation (van Schoonhoven and Pastor-Corrales) to avoid ambiguity.

Comment: Page 28, line no 414-415, check the statement and correct

Response: Statement corrected

Comments: There are some typographical and grammatical errors marked in the reviewed copy, be taken care of to improve the MS quality

Response: The typographical and grammatical errors that were highlighted in the reviewed copy have all been corrected.

Reviewer #2

Comment: There are a few typos that need due attention

Response: We have corrected all the typos that were noticed.

Comment: L98-99. Please check the sentencse -----genomic?. Position or region must be included

Response: The sentence has been corrected and physical position has also been provided.

Comment: L 104. There are about 298 races now reported in C. lindemuthianum (Nunes et al., 2021), 182 races were previously identified (Padder et al., 2017).

Response: The correction has been made and the suggested up-to-date reference (Nines et al., 2021) has been included.

Comment: L132. Delete “GWAS using”

Response: Corrected

Comment: L175. The author (year) is included but do check Plosone reference style in such a case.

Response: Corrected to be compliant with PLoS ONE reference style.

Comment: L301. Delete one that. That is written twice

Response: Corrected

Comment: L385. (Figu) or (Fig.)

Response: Corrected

Comment: L494. Banoo et al., 2020. Please see reference author (year) style of Plosone.

Response: Corrected to be compliant with PLoS ONE reference style.

---

## [Decision Letter · Decision Letter 2]

10 Oct 2023

Genome-wide association analysis of anthracnose resistance in the Yellow Bean Collection of Common Bean

PONE-D-23-01331R2

Dear Dr. Kelvin,

We’re pleased to inform you that your manuscript has been judged scientifically suitable for publication and will be formally accepted for publication once it meets all outstanding technical requirements.

Kind regards,

Karthikeyan Adhimoolam

Academic Editor

PLOS ONE

Additional Editor Comments (optional):

Reviewers' comments:

Reviewer's Responses to Questions

**Comments to the Author**

1. If the authors have adequately addressed your comments raised in a previous round of review and you feel that this manuscript is now acceptable for publication, you may indicate that here to bypass the “Comments to the Author” section, enter your conflict of interest statement in the “Confidential to Editor” section, and submit your "Accept" recommendation.

Reviewer #1: All comments have been addressed

2. Is the manuscript technically sound, and do the data support the conclusions?

Reviewer #1: Yes

3. Has the statistical analysis been performed appropriately and rigorously? 

Reviewer #1: Yes

4. Have the authors made all data underlying the findings in their manuscript fully available?

Reviewer #1: Yes

5. Is the manuscript presented in an intelligible fashion and written in standard English?

Reviewer #1: Yes

6. Review Comments to the Author

Reviewer #1: The authors have revised the MS as per the observations of the reviewers.The MS is in order now, however, assure that MS is as per journal format.

7. PLOS authors have the option to publish the peer review history of their article (what does this mean?). If published, this will include your full peer review and any attached files.

Reviewer #1: No

---

## [Editor Report · Acceptance letter]

16 Oct 2023

PONE-D-23-01331R2 

Genome-wide association analysis of anthracnose resistance in the Yellow Bean Collection of Common Bean 

Dear Dr. Kamfwa:

I'm pleased to inform you that your manuscript has been deemed suitable for publication in PLOS ONE. Congratulations! Your manuscript is now with our production department. 

Kind regards, 

on behalf of

Dr. Karthikeyan Adhimoolam 

Academic Editor

PLOS ONE